# Chemical Composition, Antioxidant, and Antimicrobial Activity of *Dracocephalum moldavica* L. Essential Oil and Hydrolate

**DOI:** 10.3390/plants11070941

**Published:** 2022-03-31

**Authors:** Milica Aćimović, Olja Šovljanski, Vanja Šeregelj, Lato Pezo, Valtcho D. Zheljazkov, Jovana Ljujić, Ana Tomić, Gordana Ćetković, Jasna Čanadanović-Brunet, Ana Miljković, Ljubodrag Vujisić

**Affiliations:** 1Institute of Field and Vegetable Crops Novi Sad, Maksima Gorkog 30, 21000 Novi Sad, Serbia; 2Faculty of Technology, University of Novi Sad, Bulevar cara Lazara 1, 21000 Novi Sad, Serbia; oljasovljanski@uns.ac.rs (O.Š.); vanjaseregelj@tf.uns.ac.rs (V.Š.); anav@uns.ac.rs (A.T.); cetkovic@tf.uns.ac.rs (G.Ć.); jasnab@uns.ac.rs (J.Č.-B.); 3Institute of General and Physical Chemistry, Studentski trg 12-16, 11000 Belgrade, Serbia; latopezo@yahoo.co.uk; 4College of Agricultural Sciences, Oregon State University, Corvallis, OR 97331, USA; valtcho.jeliazkov@oregonstate.edu; 5Faculty of Chemistry, University of Belgrade, Studentski trg 12-16, 11000 Belgrade, Serbia; jovanalj@chem.bg.ac.rs (J.L.); ljubaw@chem.bg.ac.rs (L.V.); 6Faculty of Medicine, University of Novi Sad, Hajduk Veljkova 3, 21000 Novi Sad, Serbia; ana.miljkovic@mf.uns.ac.rs

**Keywords:** Moldavian dragonhead, phytochemicals, large-scale distillation, in vitro biological activity, time-kill kinetics modeling

## Abstract

Steam distillation was used for the isolation of *Dracocephalum moldavica* L. (Moldavian dragonhead) essential oil (DMEO). This aromatic herbaceous plant is widespread across the Northern Hemisphere regions and has been utilized in health-improving studies and applications. In addition to the DMEO, the hydrolate (DMH), a byproduct of the distillation process, was also collected. The DMEO and DMH were analyzed and compared in terms of their chemical composition, as well as their in vitro biological activities. The main component in DMEO was geranyl acetate, while geranial was dominant in DMH. The DMEO demonstrated better antioxidant and antimicrobial activities compared with the DMH against *Staphylococcus aureus*, *Escherichia coli*, *Salmonella* Typhimurium, and *Listeria monocytogenes*, which represent sources of food-borne illness at the global level. The DMEO and DMH show promise as antioxidant and antimicrobial additives to various products.

## 1. Introduction

Medicinal and aromatic plants have been utilized in traditional medicine around the world for millennia and continue to play an important role in treating diseases and providing nutritive support [1]. *Dracocephalum moldavica* L., known as Moldavian dragonhead, or Moldavian balm, is an aromatic herbaceous plant native to the temperate climate of Asia, however, nowadays it is found across the Northern Hemisphere [2]. *D. moldavica* essential oil (DMEO) has a citrus-like flavor due to the contents of geranial, neral, and geranyl acetate, and reassembles other lemon-scented plants such as lemon balm and lemon catnip [3,4]. Previous research showed that this plant had antioxidant [5,6,7,8,9] and antimicrobial properties [10,11,12,13]. In addition, the plant also displayed sedative [14,15], antidepressant [16], antinociceptive [17], anti-inflammatory [18,19], as well as neuroprotective [20,21] and cardioprotective effects [22,23,24,25].

Recently, there has been an increased interest in its therapeutic benefits due to reports on its biological activities. Indeed, medicinal plants are being used in healthcare and everyday nutrition, mainly as functional food and nutraceuticals [26,27], but also as natural preservatives [28,29]. Medicinal plants can be used fresh or dry, in the form of herbal tea, extract, essential oil, or pharmaceutical formulations (tablets, capsules, etc.). Plant essential oils possess a wide spectrum of biological activities, and are used in the food industry, but also in human and veterinary medicine [30]. Hydrolates (hydrosols and distillate waters) are byproducts of steam distillation. Generally, hydrolates contain a small amount of dissolved essential oil and other constituents and possess some biological activities [31]. Therefore, hydrolates may have potential application in the food industry for flavoring, preservation, and in soft drinks, but can also be applied in aromatherapy, cosmetics, agriculture, and veterinary medicine [32].

This investigation aimed to determine the composition of essential oil and hydrolate of *D*. *moldavica* (DMEO and DMH, respectively), grown as an essential-oil-bearing crop in the Republic of Serbia and distilled under semi-industrial conditions, and the in vitro antioxidant and antimicrobial activities.

## 2. Results

### 2.1. Volatile Compounds of Essential Oil and Hydrolate

The main compounds detected in the DMEO and DMH are presented in Table 1 (GC-FID chromatograms of essential oil and hydrolate are given in the Appendix A, respectively). The most abundant compounds in the DMEO (20 compounds, comprising 98.1%) were geranyl acetate (53.2%), followed by geranial (16.8%), and neral (10.7%), while in the hydrolate, there were 23 identified compounds (comprising 96.0%), and the most abundant were geranial (23.4%), neral (22.4%), and geraniol (21.3%) (Table 1).

### 2.2. Antioxidant Activity

The results of in vitro antioxidant activity of DMEO and DMH are shown in Table 2. As a consequence of the high content of geranyl acetate, the DMEO exhibited a significantly stronger antioxidant potential than the DMH. The DMEO at concentration of 250 mg/mL showed the highest scavenging activity against lipid radicals (397.20 μmolTE/100 mL), followed by ABTS^•+^ (312.54 μmolTE/100 mL), superoxide anion (294.77 μmolTE/100 mL), and DPPH^•^ (246.39 μmolTE/100 mL). The reducing power of DMEO was lower than presented scavenging abilities but still with a high value of 171.46 μmolTE/100 mL.

### 2.3. Antimicrobial Activity

According to the obtained results of in vitro antimicrobial activity, a significant antimicrobial effect against almost all of the tested bacteria was gained, but not against the tested strains of yeasts and fungi (Table 3). The tested DMEO did not show an antimicrobial effect against *B. cereus* and *P. aeruginosa* strains, while in the case of the tested DMH, the antimicrobial effect was also absent against *S.* Typhimurium. *Pseudomonas aeruginosa* represents a very resistant strain, well-known as bacteria resistant to numerous antibiotics, and indeed, showed resistance to the antibiotics in this study.

The antimicrobial potential of DMEO and DMH toward sensitive microorganisms was conducted by the microdilution method after the satisfactory results of the preliminary screening. As shown in Table 4, the obtained MICs of DMEO showed relatively low activity against all bacteria (MIC ≤ 3.125%). Conversely, higher MICs (between 3.125–12.5%) of DMH were noted for sensitive bacteria. Consequently, the tested essential oil and hydrolate can be utilized as antimicrobial agents for the worldwide struggle for prolonged shelf-life of food, absence of food-borne pathogens, and epidemic crisis, but also can be used as an eco-based substance in antimicrobial formulations in the pharmaceutical and cosmetics industries.

The in vitro antimicrobial potential of tested substances can be further clarified by the time-kill or pharmacodynamics kinetics monitoring. In this way, in vitro examination of antimicrobial substances can be quantified in view of the path of antimicrobial activity as a function of contact time between sensitive microorganisms and targeted concentration of tested essential oil and hydrolate [33]. Therefore, a pharmacodynamics pathway of the antimicrobial effect of tested samples was conducted for all sensitive bacteria. The first step was to determine bacterial profile growth curves without the addition of either DMEO nor DMH. Non-treated bacterial suspensions were verified at the same time as the essential oil- or hydrolate-treated samples. Bacterial growth profile curves (Figure 1) indicated the number of live bacterial cells over an incubation period. There are noticeably three growth phases for all four tested bacteria: lag phase, exponential (log), and stationary phase. The initial phase is especially emphasized for *L. monocytogenes* and *S. aureus*, while this period of cell adaption is minimal for the other two sensitive bacteria. The differences are detected for maximum yield, which was obtained after approximately 12 h of incubation. Briefly, the highest concentration of bacterial cells was 7.6 CFU/mL, 8 CFU/mL, 8.4 CFU/mL, and 8.8 log CFU/mL for *S. aureus*, *L. monocytogenes*, *S*. Typhimurium, and *E. coli*, respectively. Regression coefficients for the obtained growth profile curves are shown in Table 5. Additionally, the fit between experimental and model calculated results are given in the Appendix A, indicating a very good predictive capacity of the obtained models, with a coefficient of determination of 0.99 for all tested bacteria.

The pharmacodynamics potential of DMEO at different concentrations is graphically represented in Figure 2. Kinetics profiles for MIC value indicate the biocide effect for *S. aureus* and *S.* Typhimurium after a contact time of 3 h, while the same effect was observed for *L. monocytogenes* and *E. coli* for 4 h and 12 h, respectively. A bactericidal effect was achieved in twice as short a time for *S. aureus* when 2- and 4-time MIC were applied. Interestingly, the effect of double MIC concentration did not decrease contact time which is necessary for complete inhibition of *L. monocytogenes* but quadruple MIC enabled achieving bactericidal effect after only 1 h of contact. Similar behavior was observed for *S.* Typhimurium, but with a biocide effect after 3 h and 2 h for 2- and 4-time MIC, respectively. The killing rate of 2- and 4-time MIC of DMEO for *E. coli* was achieved for 2 h or 4 h shorter contact time compared with the MIC effect.

Additionally, Table 6 summarizes regression coefficients of the kinetics models. This parameter simplifies the speed and intensity of MIC and multiple MIC values. Furthermore, Appendix A involves the goodness of fit between experimentally and model obtained results. It can be concluded that the kinetics models (Figure 2) were precise, with high coefficients of determination (0.93–1), and can be used to understand the antimicrobial effect of the DMEO against sensitive bacteria.

As previously noticed, MIC values for DMH are significantly higher compared with DMEO in the case of *L. monocytogenes* and *S. aureus*. The same concentration of DMH and DMEO is necessary for the inhibition of *E. coli* activity. Regardless of the mentioned difference, the same pharmacodynamics study was done for hydrolate-treated samples (Figure 3). The biocide effect of MIC values for all three sensitive bacteria indicated a rapid bacteriostatic effect, showing a decrease in cell viability after the first three hours of contact time. The final biocide effect was observed after 6 h. The biocide effect of MIC and double MIC values was observed in the same contact time for in the case of *L. monocytogenes* and *E. coli*. On the other hand, using a 2-time MIC value reduced the required contact time for *S. aureus*. The complete reduction in bacterial viability was detected for 4 h, 5 h, or 6 h contact time in the case of using 4-time MIC values for *L. monocytogenes*, *S. aureus*, and *E. coli*, respectively. In this study, the antimicrobial effect of DMH as a byproduct in DMEO production was reasonable suggesting new possibilities for the utilization of DMH.

According to the regression coefficients for the gained kinetics models (Table 7) and the fitness between experimentally and model obtained results (Appendix A), it can be concluded that the models were accurate, with high coefficients of determination (0.95–1) and the proposed models fit well with the experimental data. In summary, the obtained models for DMH-treated samples can be used for the prediction of antimicrobial effect based on contact time between bacterial cells and DMH.

## 3. Discussion

### 3.1. Volatile Compounds of Essential Oil and Hydrolate

According to referenced data from the literature about DMEO chemical composition (Table 8), as well as cluster analysis performed using this data for the construction an unrooted phylogenetic tree (Figure 4), it could be assumed that the largest number of accessions belonged to the geranial + neral + geranyl acetate chemotype [2,9,10,12,17,34,35,36,37,38,39,40,41,42,43,44,45,46]. The *D. moldavica* plants grown and utilized in this study also belonged to the above chemotype. Other previously reported chemotypes of *D. moldavica* include: the geranyl acetate + geranial + geraniol one [42,47,48,49,50,51], the 1,8 cineole + 4-terpineol [52], and the linalool + geranial + fenchone [53].

The phylogenetic cluster tree for DMEO was estimated and drawn using R software 4.0.3, as it is previously described for essential oil of white horehound [54], immortelle [55], hyssop [56], naked catmint [57], and sweet wormwood [58].

### 3.2. Antioxidant Activity

Plant essential oils represent natural sources of bioactive compounds with several models of action, among them, scavenging of free radicals, prevention of chain reactions initiation, reducing agents, and termination of peroxides, quenching of singlet oxygen, and binding of metal ion catalysts [59]. Different in vitro assays have been used to estimate the antioxidant potential of DMEO, while the data of DMH antioxidant activity have not been previously published in the scientific literature. In general, for complex systems such as essential oils, using at least three methods with different mechanisms is preferable and in some cases required to evaluate antioxidant activities. The DPPH^•^ and ABTS^•+^ assays are efficient tools for the determination of antioxidant activity; although these methods have an identical mechanism, the mediums for the assay are different which means there is a dissimilar solubility of the isolated bioactive compounds [60]. The SOA, like ABTS^•+^ assay, share the water medium, but at a different pH level. Superoxide anion is the most frequent free radical in vivo and deals as a precursor for other reactive oxygen species that possess the capability to induce damage of important biological molecules. In the BCB assay, due to the absence of antioxidants, *β*-carotene undergoes rapid discoloration; this could be explained by the integrated oxidation of *β*-carotene and linoleic acid-generating free lipid species. Another defense mechanism in preventing the body from the dangerous effect of free radicals is reducing these molecules by the antioxidants. In the present study, reducing power was monitored through measurement of the ferrous ions transformation in presence of antioxidants.

Geranyl acetate is known for its strong antioxidant properties due to its capacity to reduce free radical stability via electron or hydrogen donating mechanisms [61]. As well, geranyl acetate is insoluble in water but soluble in organic solvents and oil, which is the reason for its low concentration in DMH.

Investigation of the DMEO antioxidant capacity obtained from *D. moldavica* plants grown as a single-crop vs. intercropping systems with soybean in response to the application of chemical fertilizer (urea and triple superphosphate) and organic manure had the IC50 values in the range from 1.45 to 5.28 μg/mL [9]. Furthermore, the evaluation of the antioxidant activity of DMEO using DPPH^•^, ABTS^•+^, and BCB assays showed that essential oil possesses weaker scavenger activity for DPPH^•^ and ABTS^•+^ radicals than ascorbic acid and BHT, while higher activity was reported for peroxyl radicals [12]. The results from the latter study were in agreement with the results of the present study. Generally, both DMEO and DMH contain very efficient bioactive compounds such as geranyl acetate, geranial, geraniol, and neral, which are responsible for the antioxidant activity. There is not enough information about antioxidant mechanisms and other biological activities of DMEO and DMH.

### 3.3. Antimicrobial Activity

The gained antimicrobial effect (see Section 2.3) can be the result of the chemical composition of the DMEO and DMH, due to the fact that geranyl acetate, as an ester derived from geraniol, as well as geranial and neral (together known as citral) have good antibacterial properties and good thermal stability [62,63]. The mentioned group of bioactives as well as this group of bioactives are especially dominant in DMEO, but also in DMH (Table 2).

Resistance of *B. cereus* on the tested samples was not in line with the results of El-Baky and El-Baroty [11] that reported the inhibitory effect of DMEO from Egypt in a concentration of 0.07 mg/mL. This may suggest the difference in antimicrobial effect based on the origin of the plant and differences in its chemical profile, which has been reported previously [64,65,66]. Both DMEO and DMH samples in this study showed an antimicrobial effect on *S. aureus*, *L. monocytogenes*, and *E. coli* strains that is strongly correlated to the results of Eshani et al. [12], which demonstrated a significant antimicrobial effect of DMEO against the mentioned bacteria. Notably, the obtained max of 40 mm was registered for all the tested bacteria in the case of DMEO, while this value for the tested DMH was lower. Moreover, the inhibition zones of both DMEO and DMH were even higher than that in the positive control, i.e., cefotaxime and clavulanic acid combination, indicating the significant potential of using the DMEO and DMH as natural ingredients in various products to improve microbial antibiotic resistance. To the best of our knowledge, the high antimicrobial performance of DMH has not been previously reported. In addition to the antibacterial effect, several scientific groups reported antifungal activity of DMEO [10,67,68] which did not correlate with the results of this research.

Additional observation can be directed to the fact that all four sensitive bacteria represent common pathogenic bacteria that cause foodborne diseases [69], while alimentary infection and intoxications caused by these pathogens represent a growing public health problem [70]. Due to the mentioned facts, the obtained results are promising in view of finding alternative agents with rapid biocide effect for the food or packaging production.

## 4. Materials and Methods

### 4.1. Plant Material

*D. moldavica* was grown during the 2021 cropping season in the Institute of Field and Vegetable Crops Novi Sad (IFVCNS), experimental fields in Bački Petrovac (Vojvodina Province, North Serbia) on *gleyed calcareous chernozem* soil type. The species were confirmed by Milica Rat, research associate at the botanical collection, and deposited under the voucher number 2-1513 at the Herbarium BUNS (Faculty of Sciences, University of Novi Sad). The previous crop was barley. Granular mineral fertilizer (70 kg NPK in formulation 15:15:15) was applied in the previous fall prior to the fall plowing and disking.

The seeds were sown in pots in a greenhouse, in March, and the seedlings were transplanted at the end of April in an experimental plot of 70 m × 10 m, with a 70 cm spacing between rows and 50 cm between plants. During the vegetation period, only hand weeding and hoeing were performed. Plants were harvested in August, at full bloom, by cutting the plants at 5 cm above the ground, dried in a flat-bed solar dryer at temperatures less than 40 °C for two days, and the essential oil was isolated via steam distillation.

### 4.2. Essential Oil Isolation

The steam distillation of the dried aerial plant parts of *D. moldavica* was performed in a semi-industrial distillation unit at IFVCNS [71]. Briefly, 50 kg of dried biomass was put in the distillation vessel (0.8 m^3^), which was supplied with hot dry steam from a separate steam generator. After 20 min, a condensate (essential oil and condensed water) started to accumulate in the glass Florentine flask. After 4 h of distillation, the essential oil and the hydrolate were separated: the DMEO was decanted from the aqueous layer, dried over anhydrous sodium sulfate, while the DMH was purified by filtration using MN 651/120 filter paper. The essential oil yield was 0.65% in dried biomass. In order to prepare the DMH sample for analysis of volatile compounds, 400 mL of hydrolate were extracted by dichloromethane via a Likens–Nickerson apparatus for 2 h.

### 4.3. Analysis of Volatile Compounds

Gas chromatograph (Agilent 7890A) with two detectors, flame ionization (FID) and mass selective (Agilent 5975C)*,* and non-polar capillary column HP-5MS (30 m × 0.25 mm × 0.25 μm) were used for the analysis of DMEO and DMH. The operating conditions were the same as in our previous works [58,71]. Identification of the components was conducted according to their linear retention indices (RI), and comparison with mass spectral libraries (Adams ver. 4, Wiley ver. 5, and NIST ver. 17). The relative abundance of each detected compound was calculated from GC/FID chromatograms as a percentage area of each peak (only identified compounds are shown).

### 4.4. In vitro Assessment of Antioxidant Activity

The potential antioxidant activity of DMEO and DMH were assessed using five common in vitro antioxidant assays. The tests were performed with DMEO dissolved in methanol at the concentration of 250 mg/mL. For all assays, the Trolox equivalents were used for expression of antioxidant activities as μmol per 100 mL (μmolTE/100 mL). The DMH was tested in its original eluted form after filtration.

#### 4.4.1. DPPH^•^ (2,2-diphenyl-1-picrylhydrazyl)

The DPPH assay was performed according to Aborus et al. [72]. Briefly, 250 μL DPPH^●^ solution in methanol (0.89 mM) was mixed with 10 μL of the sample in a microplate well and left in the dark at ambient temperature for 50 min. Absorbance was measured at 515 nm, and methanol was used as a blank.

#### 4.4.2. ABTS^•+^ (2,2′-azino-bis-3-ethylbenzothiazoline-6-sulphonic Acid)

The ABTS^•+^ radical scavenging assay was evaluated employing the method according to Aborus et al. [72]. The absorbances of 250 μL activated ABTS^•+^ (with MnO_2_), before and 35 min (incubated at 25 °C) after the addition of 2 μL of juice were measured at 414 nm. Water was used as blank.

#### 4.4.3. SOA (Superoxide Anion)

Superoxide anion radical scavenging activity was determined by the nitroblue tetrazolium reduction method, adapted for a 96-well microplate [73]. An amount of 50 µL of each sample solution dissolved in phosphate buffer, or 50 µL of buffer (blank test), was mixed with 50 µL of 166 mM nicotinamide adenine dinucleotide (NADH), 150 µL of 43 mM nitrotetrazolium blue (NBT), and 50 µL of phenazine methosulfate (PMS) in triplicate. The tests were conducted at room temperature with two readings of 560 nm, being the initial when PMS is added, and the final after 2 min.

#### 4.4.4. RP (Reducing Power)

Reducing power (RP) was determined by the method of Oyaizu [74] adapted for a 96-well microplate. In brief, a 25 μL sample or 25 μL water (blank test), 25 μL sodium phosphate buffer (pH = 6.6), and 25 μL of 1% potassium iron(III) cyanide were mixed and incubated in a water bath for 20 min at 50 °C. After cooling, 25 μL of 10% trichloroacetic acid was added and solutions were centrifuged at 2470× *g* for 10 min. After centrifugation, 50 μL of supernatant was mixed with 50 μL of distilled water and 10 μL of 0.1% iron(III) chloride in the microplate. Absorbances were measured immediately at 700 nm.

#### 4.4.5. BCB (*β*-Carotene Bleaching)

The *β*-carotene bleaching capacity of samples was evaluated by the *β*-carotene linoleate model system of Al-Saikhan et al. [75]. The absorbances of all the samples were taken at 470 nm at zero time and after 180 min, while during this time the microplate was incubated at 50 °C.

### 4.5. In Vitro Assessment of the Antimicrobial Activity

Observation of the antimicrobial activity of the DMEO and DMH was performed using references strains of bacteria *Bacillus cereus*, *Escherichia coli*, *Listeria monocytogenes*, *Pseudomonas aeruginosa*, *Salmonella* Typhimurium, and *Staphylococcus aureus*, as well as referent representatives of yeasts and fungi (*Aspergillus brasiliensis*, *Candida albicans*, and *Saccharomyces cerevisiae*). All tested strains were obtained from the American Type Culture Collection (ATCC).

#### 4.5.1. Screening of Antimicrobial Effect of DMEO and DMH (Disk Diffusion Method)

Evaluation of antimicrobial activity of the DMEO and DMH was completed by the disk-diffusion method. As Micić et al. [76] reported in detail, the nutrient medium (Müller–Hinton agar or Sobouraud maltose agar) was inoculated with microbial suspensions (approx. 6 log CFU/mL) and the samples (15 μL) were applied onto three sterile cellulose discs. Bacteria were grown on Müller–Hinton agar (HiMedia, Mumbai, India) at 37 °C for 24 h and at 30 °C (*B. cereus*) for 18 h. Yeast and fungi were grown on Sabouraud maltose agar (HiMedia, Mumbai, India) at 25 °C for 48 h. Cells were suspended in a sterile 0.9% NaCl solution. As a negative control, sterile distilled water was used, while positive controls were commercially available antibiotics chloramphenicol and tetracycline (Sigma-Aldrich, St. Louis, MO, USA) as well as actidione (Biochemica, Billingham, U.K.). The obtained results were interpreted as follows: sensitive (diameter of inhibition zone above 26 mm), intermediary (inhibition zone 22–26 mm), and resistance (inhibition zone below 22 mm).

#### 4.5.2. Minimal Inhibitory Concentration (MIC)

The MIC was evaluated for all bacteria, yeast, and fungi that are sensitive to the DMEO and DMH using the microdilution method labeled by Pavlić et al. [77]. The initial concentration was defined as 100%, while other concentrations were prepared using successive dilutions (100–0.39%) using dimethyl sulfoxide (50 mg/mL) for essential oil or sterile distilled water for hydrolate. The used dissolvents were inert for all tested microorganisms and did not have a biocide effect on bacterial and yeast growth [66]. MIC represents the lowest concentration of antimicrobial agents that, under defined in vitro conditions, prevents the appearance of visible growth of a microorganism within a defined period of time. MIC is calculated based on the numbers of cells of positive control and treated samples with DMEO or DMH. The test microtiter plate had a positive control (inoculated media without DMEO or DMH) and a negative control (100 µL of medium mixed with 100 µL of DMEO or DMH).

#### 4.5.3. Assessment of Antimicrobial Activity Using a Time-Kill Procedure

The pharmacodynamics potential of antimicrobial activity was followed using monitoring of the time-kill kinetics as reported by Ferro et al. [78]. All sensitive bacteria (approx. 6 log CFU/mL) were tested during contact time with 1, 2, and 4-time MIC concentrations in several samples times (0 h, 2 h, 3 h, 4 h, 5 h, 6 h, 12 h, and 24 h of incubation for bacteria at 37 °C). An inoculated medium without the sample was positive control, while a non-inoculated medium was a blank. The four-parameter sigmoidal model established by Romano et al. [79] performed kinetic modeling.

### 4.6. Statistical Analyses

The collected data were processed statistically using the software package STATISTICA 10.0 (StatSoft Inc., Tulsa, OK, USA). All analyses were performed in three replicates. The obtained results were expressed as the mean value with standard deviation (SD). Analysis of variance (ANOVA) with Tukey’s HSD post hoc test for comparison of the sample means were used to explore the variations of parameters. All observed samples were checked for variance equality (using Levene’s test) and normal distribution (using Shapiro–Wilk’s test).

## 5. Conclusions

Expressed needs for natural materials and phytochemicals, as well as needs to find antimicrobial substances that are substitutes for antibiotics, inspire researchers to look for new sources of these compounds. Herbs, such as *D. moldavica*, could be important raw materials in the pharmaceutical and food industry. The main compounds in the essential oil were geranyl acetate, geranial, and neral, while in the hydrolate these were geranial, neral, and geraniol. Geranyl acetate, a monoterpene, is commonly used at an industrial level in a wide range of products such as powders, soaps, perfumes, as well as flavoring agents, due to its intensely fruity and floral aroma. Due to its low solubility in water, it was not detected in the hydrolate. Citral (a mixture of two monoterpene aldehydes—geranial and neral) is widely used as a flavoring agent in food, beverage, and cosmetic products. *Geraniol* is a commercially important terpene alcohol used in the food, fragrance, and cosmetic industry, as well as insecticidal and repellent compounds in pesticides and household products. *D. moldavica* essential oil (DMEO) and hydrolate (DMH) as significant sources of these compounds are prospective raw materials for many purposes.

## Figures and Tables

**Figure 1 plants-11-00941-f001:**
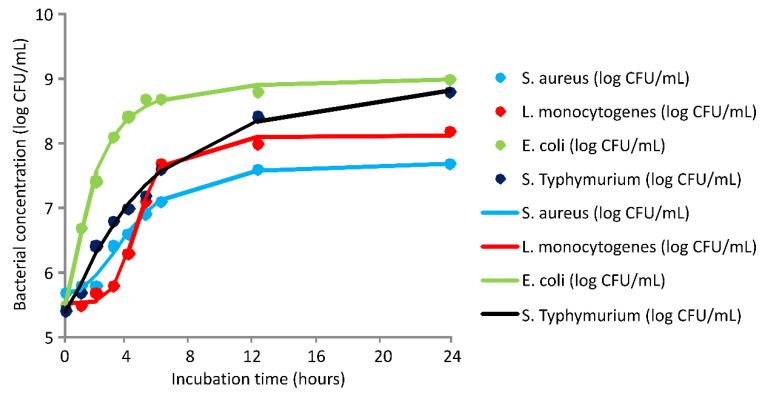
Bacterial growth kinetics—control samples (incubation of sensitive bacteria without the tested antimicrobial substances)—markers signify the experimental data; lines indicate predictive results.

**Figure 2 plants-11-00941-f002:**
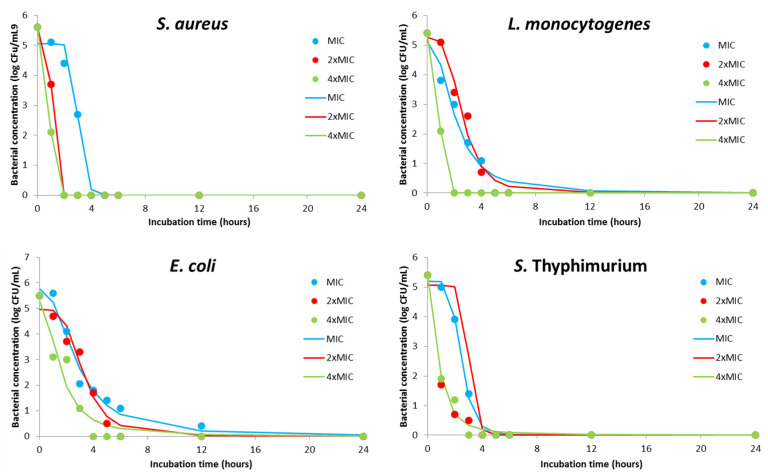
Kinetics modeling for pharmacodynamics potential of antimicrobial activity using MIC, 2×MIC, and 4×MIC of DMEO (markers signify the experimental data; lines indicate predictive results).

**Figure 3 plants-11-00941-f003:**
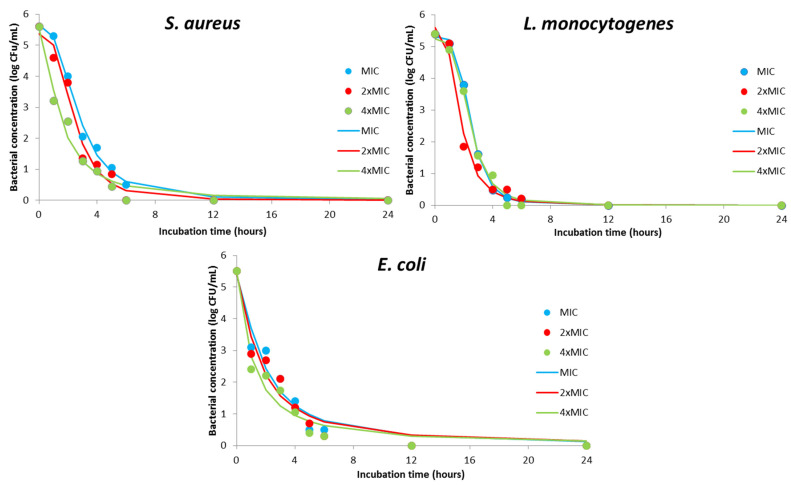
Kinetics modeling for pharmacodynamics potential of antimicrobial activity using MIC, 2×MIC, and 4×MIC of DMH (markers signify the experimental data; lines indicate predictive results).

**Figure 4 plants-11-00941-f004:**
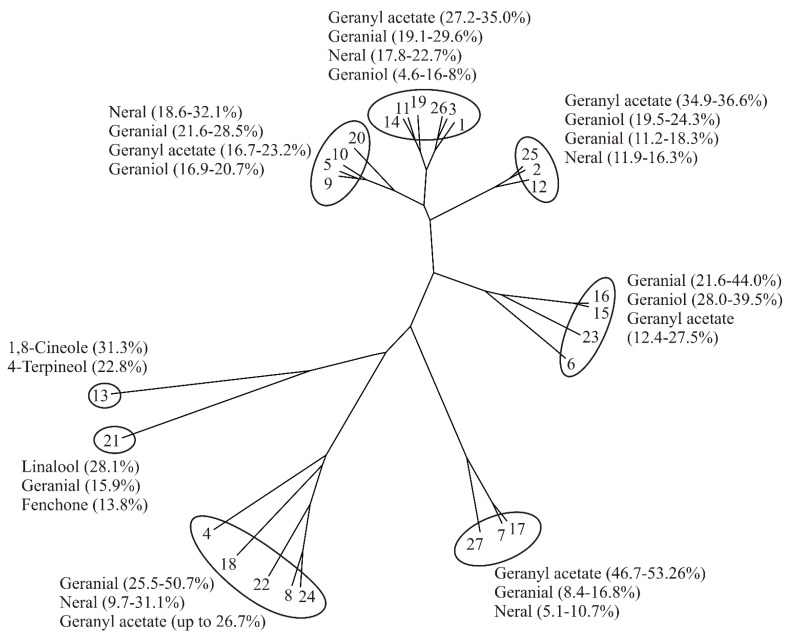
The unrooted phylogenetic tree of *D. moldavica* essential oil (DMEO) according to Table 8.

**Table 1 plants-11-00941-t001:** Chemical composition of *D. moldavica* essential oil (DMEO) and hydrolate (DMH).

No	Compound	RI	DMEO	DMH
1	1-octen-3-ol	974	-	0.5
2	6-methyl-5-hepten-2-one	986	0.1	3.3
3	dehydro-1,8-cineole	988	-	1.3
4	3-octanol	995	-	0.2
5	1,8-cineole	1028	-	0.6
6	Benzene acetaldehyde	1041	-	0.6
7	*cis*-linalool oxide (furanoid)	1069	-	1.7
8	*trans*-linalool oxide (furanoid)	1086	-	0.7
9	Linalool	1096	1.6	8.7
10	Camphor	1141	-	0.5
11	*trans*-chrysanthemal	1147	-	0.3
12	Nerol oxide	1149	0.1	0.4
13	Borneol	1159	0.6	2.9
14	Terpinen-4-ol	1171	0.1	1.7
15	Cryptone	1185	-	0.1
16	*α*-Terpineol	1188	-	0.9
17	*trans*-Isocitral	1777	0.4	-
18	Nerol	1227	-	2.2
19	Neral	1234	10.7	22.4
20	Linalool acetate	1247	7.9	-
21	Geraniol	1254	-	21.3
22	Geranial	1266	16.8	23.4
23	Lavandulyl acetate	1285	0.1	-
24	Methyl geranate	1318	0.1	-
25	Eugenol	1356	-	0.5
26	Neryl acetate	1358	4.6	0.1
27	*α*-Copaene	1369	0.2	-
28	Geranyl acetate	1379	53.2	1.7
29	*trans*-caryophyllene	1412	0.6	-
30	*α*-humulene	1447	0.1	-
31	*trans*-*β*-farnesene	1451	0.1	-
32	*γ*-muurolene	1474	0.4	-
33	E,E-*α*-farnesene	1502	0.1	-
34	Caryophyllene oxide	1575	0.3	-
	Total		98.1	96.0

RI—Retention Indices on non-polar capillary column HP-5MS.

**Table 2 plants-11-00941-t002:** In vitro antioxidant activity of *D. moldavica* essential oil (DMEO) and hydrolate (DMH).

Antioxidant Activity (μmolTE/100 mL)	DMEO	DMH
DPPH^•^	246.39 ± 1.17 ^b^	8.82 ± 0.36 ^a^
ABTS^•+^	312.54 ± 11.63 ^b^	25.44 ± 1.98 ^a^
SOA	294.77 ± 13.29 ^b^	19.58 ± 0.11 ^a^
BCB	397.20 ± 36.12 ^b^	41.63 ± 2.17 ^a^
RP	171.46 ± 2.56 ^b^	9.50 ± 0.64 ^a^

Results are expressed as mean ± standard deviation (*n* = 3). Values in the row with different superscripts are significantly different at *p* < 0.05 according to Fisher’s least significant differences (LSD) test. DPPH^•^—2,2-diphenyl-1-picrylhydrazyl; ABTS^•+^—2,2′-azino-bis-3-ethylbenzothiazoline-6-sulphonic acid; SOA—superoxide anion; BCB—*β*-carotene bleaching; and RP—reducing power.

**Table 3 plants-11-00941-t003:** Assessment of the antimicrobial effect.

Test Organisms	The Inhibition Zone * (mm)			
	*D. moldavica*		Controls	
	DMEO	DMH	Antibiotic	Actidione
*B*. *cereus* ATCC 11778	16.0 ± 0.0	11.0 ± 0.0	27.0 ± 0.0	-
*S*. *aureus* ATCC 25923	40.0 ± 0.0	28.33 ± 0.0	28.0 ± 0.0	-
*L*. *monocytogenes* ATCC 35152	40.0 ± 0.0	27.0 ± 0.0	26.3 ± 0.6	-
*E*. *coli* ATCC 25922	40.0 ± 0.0	34.33 ± 0.58	27.0 ± 0.0	-
*P*. *aeruginosa* ATCC 27853	21.0 ± 0.0	nd	21.0 ± 0.0	-
*S*. Typhimurium ATCC 13311	40.0 ± 0.0	10.33 ± 0.58	29.33 ± 0.6	-
*S*. *cerevisiae* ATCC 9763	11.33 ± 0.58	nd	-	34.0 ± 0.0
*C*. *albicans* ATCC 10231	nd	nd	-	37.0 ± 0.0
*A*. *brasiliensis* ATCC 16404	nd	nd	-	27.3 ± 0.6

* Mean value diameter of zone including disc (6 mm) ± standard deviation; nd—not detected.

**Table 4 plants-11-00941-t004:** Minimal inhibitory concentration (%).

Test Organisms	DMEO	DMH
*B*. *cereus* ATCC 11778	>100 *	>100
*S*. *aureus* ATCC 25923	0.78	12.5
*L*. *monocytogenes* ATCC 35152	1.56	6.25
*E*. *coli* ATCC 25922	3.125	3.215
*P*. *aeruginosa* ATCC 27853	>100	>100
*S*. Typhimurium ATCC 13311	0.78	>100
*S*. *cerevisiae* ATCC 9763	>100	>100
*C*. *albicans* ATCC 10231	>100	>100
*A*. *brasiliensis* ATCC 16404	>100	>100

* According resistance on the initial concentration (see Table 3).

**Table 5 plants-11-00941-t005:** Regression coefficients for bacterial growth kinetics (control samples).

Coefficient	Bacterial Concentration (Log CFU/mL)			
	*S. aureus*	*L. monocytogenes*	*E. coli*	*S.* Typhimurium
d	7.72	8.12	9.04	9.27
a	5.7	5.53	5.52	5.38
c	4.29	4.60	1.64	4.95
b	2.58	5.59	1.65	1.30

The regression coefficients could be depicted as follows: a–minimum of the experimentally gained values (t = 0); d–the maximally gained values (t = ∞); c–the infection point (the point between a and d); b–the steepness of the infection point c.

**Table 6 plants-11-00941-t006:** Regression coefficients for time-kill kinetics models for DMEO-treated samples.

Coefficients	DMEO Concentration											
	*S. aureus*			*L. monocytogenes*			*E. coli*			*S.* Typhimurium		
	MIC	2×MIC	4×MIC	MIC	2×MIC	4×MIC	MIC	2×MIC	4×MIC	MIC	2×MIC	4×MIC
d	0.00	0.00	0.00	0.00	0.00	0.00	0.00	0.00	0.00	0.00	0.00	0.00
a	5.05	5.6	5.6	5.13	5.27	5.4	5.77	4.97	5.31	5.20	5.05	5.39
c	3.03	1.05	0.96	2.06	2.59	0.96	2.78	3.25	1.53	2.45	3.03	0.77
b	11.64	14.9	11.7	2.34	3.65	11.89	2.26	3.9	2.06	5.72	11.64	1.99

The regression coefficients could be depicted as follows: a–minimum of the experimentally gained values (t = 0); d–the maximally gained values (t = ∞); c–the infection point (the point between a and d); b–the steepness of the infection point c.

**Table 7 plants-11-00941-t007:** Regression coefficients for time-kill kinetics models for DMH-treated samples.

Coefficients	DMH Concentration											
	*S. aureus*			*L. monocytogenes*			*E. coli*			*S. aureus*		
	MIC	2×MIC	4×MIC	MIC	2×MIC	4×MIC	MIC	2×MIC	4×MIC	MIC	2×MIC	4×MIC
d	0.00	0.00	0.00	0.00	0.00	0.00	0.00	0.00	0.00	0.00	0.00	0.00
a	5.65	5.36	5.53	5.31	5.59	5.25	5.36	5.4	5.46	5.65	5.36	5.53
c	2.68	2.40	1.43	2.47	1.76	2.43	1.76	1.52	1.05	2.68	2.40	1.43
b	2.65	3.01	1.65	4.35	3.05	3.85	1.44	1.32	1.16	2.65	3.01	1.65

The regression coefficients could be depicted as follows: a–minimum of the experimentally gained values (t = 0); d–the maximally gained values (t = ∞); c–the infection point (the point between a and d); b–the steepness of the infection point c.

**Table 8 plants-11-00941-t008:** Chemical composition of DMEO according to the literature data.

No	Reference	1,8-Cineole	4-Terpineol	Fenchone	Geranial	Geraniol	Geranyl Acetate	Linalool	Methyl Chavicol	Neral	Nerol	Neryl Acetate
1	[2]	0.0	0.0	0.0	29.6	5.4	27.2	0.4	0.0	19.4	0.4	3.0
2	[34] *	0.3	0.0	0.1	16.3	22.3	35.6	0.3	0.0	11.9	1.0	2.6
3	[35] *	0.0	0.0	0.0	26.2	4.6	35.0	0.2	0.0	20.7	0.0	4.1
4	[36]	0.0	0.0	0.0	25.5	0.5	15.2	1.3	16.0	9.7	0.3	1.2
5	[37]	0.0	0.0	0.0	27.3	20.7	23.2	0.8	0.0	18.6	0.0	2.1
6	[47]	0.0	0.0	0.0	21.6	39.5	12.4	0.8	0.0	17.1	1.5	1.6
7	[38] *	0.0	0.0	0.0	9.3	16.0	52.7	0.6	0.0	5.1	0.3	2.9
8	[9] *	0.0	0.0	0.0	36.6	2.9	26.7	0.3	0.0	25.7	0.1	1.2
9	[39]	0.0	0.0	0.0	26.3	16.9	22.5	1.5	0.0	21.3	1.0	0.4
10	[12]	0.0	0.0	0.0	28.5	19.6	16.7	0.8	0.0	21.2	1.9	1.8
11	[40] *	0.0	0.0	0.0	23.6	16.8	29.2	2.0	0.0	20.2	1.9	0.0
12	[48]	0.0	0.0	0.0	11.2	24.3	36.6	0.8	0.0	16.3	0.4	0.9
13	[52]	31.3	22.8	0.0	0.0	0.0	0.0	0.0	0.0	0.0	0.0	0.0
14	[41]	0.0	0.0	0.0	19.8	15.1	27.9	2.4	0.0	18.0	2.2	4.2
15	[42]	0.0	0.0	0.0	30.9	34.2	25.4	0.6	0.0	0.3	0.0	1.5
16	[42]	0.0	0.0	0.0	27.8	36.0	27.5	1.0	0.0	0.4	0.0	1.8
17	[43]	0.0	0.0	0.0	8.4	15.9	46.7	0.5	0.0	5.8	0.3	2.6
18	[17]	0.0	0.0	0.0	31.1	0.0	0.0	1.5	0.0	31.1	17.1	4.8
19	[44]	0.0	0.0	0.0	19.1	9.3	30.4	2.7	0.0	17.8	2.9	2.5
20	[10]	0.0	0.0	0.0	21.6	17.6	19.9	1.1	0.0	32.1	0.0	1.6
21	[53] *	0.4	0.0	13.8	15.9	6.9	1.3	28.1	1.2	0.0	1.4	0.9
22	[45] *	0.01	0.0	0.0	50.7	3.4	10.0	0.1	0.0	26.8	0.0	0.0
23	[49]	0.0	0.0	0.0	44.0	28.0	14.0	0.4	0.0	6.3	0.2	2.6
24	[46]	0.3	0.0	0.1	41.9	5.3	19.0	0.5	0.0	25.3	0.3	0.6
25	[50]	0.0	0.0	0.0	18.3	19.5	34.9	0.4	0.0	14.8	0.0	2.9
26	[51]	0.0	0.0	0.0	24.5	8.8	32.6	0.8	0.0	22.7	2.4	3.4
27	TS	0.0	0.1	0.0	16.8	0.0	53.2	1.6	0.0	10.7	0.0	4.6

* Average value from different agrotechnical measures (cropping patterns, fertilization, and irrigation); TS—this study.

## Data Availability

The data is contained in the article and Appendix A.

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
