# Peer review of "Chemical Composition, Antioxidant, and Antimicrobial Activity of Dracocephalum moldavica L. Essential Oil and Hydrolate"

_plants, 2022, doi:10.3390/plants11070941_

Round 1

Reviewer 1 Report

The article entitled „Chemical composition, antioxidant and antimicrobial activity of Dracocephalum moldavica L. essential oil and hydrolate” describes the composition and biological activity of the essential oil and the hydrolate obtained from Dracocephalum moldavica L. To the best of my knowledge this is the first report which describe this plant in European cultivation. According to the SciFinder database, there are a dozen articles on the composition of the essential oil obtained from this plant, but from Asia. Due to this fact, I consider the research interesting and worth publishing. In my opinion, the article is clearly written and could be published after small corrections:

  1. Line 20 and 292 – use isolation instead of extraction
  2. Line 227 and 290 – use isolated instead of extracted
  3. Line 57 – add word oil after essential

Author Response

The authors would like to thank the Reviewer for professional and helpful comments. We understand that the isolation/isolated is better word than extraction/extracted, so we corrected through manuscript. Also, we added missing word (oil) as you suggested.

Reviewer 2 Report

Overall the manuscript is ok. The author needs to address the following issues.

  1. The methods should be described in detail in the methodology section.
  2. The author should provide images of inhibition zones of the antimicrobial activity study.
  3. The author needs to provide the GC spectra as a figure.
  4. All the data should be statistically significant and the author needs to carry out the statistical analysis study.
  5. The discussion should be improved with the most recent references.

Author Response

The authors would like to thank the Reviewer for professional and helpful comments. 

Reviewer 3 Report

The manuscript entitle: Chemical composition, antioxidant and antimicrobial activity of Dracocephalum moldavica L. essential oil and hydrolate have been reviewed.

The attributes to the scientific manuscript are very clear and the scientific soundness of the experimental work is signifficant.

There are only technical suggestions to improve the manuscript.

1) There is not correct numbering of the Tables and Figures - Table 2 and Figure 1 are in Discussion behind the Restuls - The higher numberst of tables and figures are front.

2) Table 3 - Add as a note under the table full expresion of SOA, BSB and RP.

Best wishes

Author Response

(The authors gave the same response as above.)

Round 2

Reviewer 2 Report

The author has addressed all the suggested comments.